# Mixed evidence for the relationship between periodontitis and Alzheimer's disease: A bidirectional Mendelian randomization study

Yi-Qian Sun[1,2]*, Rebecca C. Richmond[3], Yue Chen[4], Xiao-Mei Mai[5]

**1** Center for Oral Health Services and Research Mid-Norway (TkMidt), Trondheim, Norway, **2** Department of Clinical and Molecular Medicine (IKOM), NTNU—Norwegian University of Science and Technology, Trondheim, Norway, **3** School of Social and Community Medicine, University of Bristol, Bristol, United Kingdom, **4** School of Epidemiology and Public Health, Faculty of Medicine, University of Ottawa, Ottawa, Canada, **5** Department of Public Health and Nursing, NTNU—Norwegian University of Science and Technology, Trondheim, Norway

* yiqsu@tkmidt.no, yi-qian.sun@ntnu.no

**Data Availability Statement:** All relevant data are within the paper and its Supporting Information files.

## Abstract

Recent experimental studies indicated that a periodontitis-causing bacterium might be a causal factor for Alzheimer's disease (AD). We applied a two-sample Mendelian randomization (MR) approach to examine the potential causal relationship between chronic periodontitis and AD bidirectionally in the population of European ancestry. We used publicly available data of genome-wide association studies (GWAS) on periodontitis and AD. Five single-nucleotide polymorphisms (SNPs) were used as instrumental variables for periodontitis. For the MR analysis of periodontitis on risk of AD, the causal odds ratio (OR) and 95% confidence interval (CI) were derived from the GWAS of periodontitis (4,924 cases vs. 7,301 controls) and from the GWAS of AD (21,982 cases vs. 41,944 controls). Seven non-overlapping SNPs from another latest GWAS of periodontitis was used to validate the above association. Twenty SNPs were used as instrumental variables for AD. For the MR analysis of liability to AD on risk of periodontitis, the causal OR was derived from the GWAS of AD including 30,344 cases and 52,427 controls and from the GWAS of periodontitis consisted of 12,289 cases and 22,326 controls. We employed multiple methods of MR. Using the five SNPs as instruments of periodontitis, there was suggestive evidence of genetically predicted periodontitis being associated with a higher risk of AD (OR 1.10, 95% CI 1.02 to 1.19, $P = 0.02$). However, this association was not verified using the seven independent SNPs (OR 0.97, 95% CI 0.87 to 1.08, $P = 0.59$). There was no association of genetically predicted AD with the risk of periodontitis (OR 1.00, 95% CI 0.96 to 1.04, $P = 0.85$). In summary, we did not find convincing evidence to support periodontitis being a causal factor for the development of AD. There was also limited evidence to suggest genetic liability to AD being associated with the risk of periodontitis.

**Funding:** YQS's research is supported by funding from The Norwegian Cancer Society (project ID 5769155-2015) and The Research Council of Norway "Gaveforsterkning". RCR is a de Pass VC Research Fellow at the University of Bristol. The funders had no role in study design, data collection and analysis, decision to publish, or preparation of the manuscript.

**Competing interests:** The authors have declared that no competing interests exist.

## Introduction

Genome-wide association studies (GWAS) have identified more than 20 loci that affect the risk of Alzheimer's disease (AD) [1–3]. However, it remains elusive which environmental factors increase the risk of late onset AD. Lower education attainment and vitamin D levels have been suggested to be causally associated with the risk of AD [4–6]. Recent studies showed that *Porphyromonas gingivalis*, a keystone pathogen in chronic periodontitis, might be a causal factor for AD [7, 8]. *P. gingivalis* DNA was identified in the brain of AD patients, and oral *P. gingivalis* infection in mice resulted in brain colonization of the bacteria and increased production of amyloid-β [7].

However, evidence of a causal link between periodontitis and AD risk is limited in humans due to issues of confounding and reverse causation in the epidemiological studies [9]. The Mendelian randomization (MR) approach attempts to overcome these limitations of observational epidemiology with the use of genetic variants that serve as instrumental variables for the exposure of interest [10, 11]. Genetic variants used as instrumental variables in an MR study may be used to infer causal effect of the exposure if they satisfy three fundamental assumptions: 1) strongly associated with the exposure; 2) independent of confounding factors of the observational association; and 3) associated with the outcome only via the exposure (no horizontal pleiotropy) [10, 11]. The increase in publicly accessible summary statistics of GWAS facilitates the application of this method. In this study, we aimed to investigate the potential causal relationship between chronic periodontitis and AD in the population of European ancestry using a bidirectional two-sample MR method.

## Materials and methods

### MR of chronic periodontitis on risk of AD

Summary statistics of chronic periodontitis (S1 Table) were retrieved from a meta-analysis of GWAS of periodontitis by Munz *et al.* including 12,225 individuals (4,924 cases vs. 7,301 controls) of European ancestry [12]. Five single-nucleotide polymorphisms (SNPs) were suggestively associated with periodontitis based on $P$ value $5\times10^{-6}$ and were used as instrumental variables. Another newly published independent GWAS of periodontitis by Shungin *et al.* [13] was used for a validation analysis, in which eight SNPs (non-overlapping with the above-mentioned five SNPs) were suggestively ($P$ value $<5\times10^{-6}$) associated with periodontitis in people with European ancestry in the GLIDE (Gene-Lifestyle Interactions in Dental Endpoints) consortium (12,289 clinically diagnosed periodontitis cases vs. 22,326 controls). Summary statistics for seven of the eight SNPs were available in the GWAS of AD, and so were included as instrumental variables for periodontitis in the validation analysis (S1 Table). In this study, instrumental SNPs in linkage disequilibrium were pruned with a clumping $R^2$ cut-off 0.001 and the SNP with the lowest $P$ value was retained. Summary statistics of AD were from Stage 1 of a latest GWAS by Kunkle *et al.* (21,982 cases vs. 41,944 controls) [2], from which the effect sizes for SNPs of periodontitis were extracted. Summary statistics of Stage 1 but not the overall analysis contain the information on these SNPs. The study by Kunkle *et al.* [2] is so far the largest GWAS of clinically diagnosed AD in the population of European ancestry with publicly accessible summary statistics.

### MR of AD on risk of chronic periodontitis

Summary statistics of AD (S2 Table) were from the analysis of overall stages 1 and 2 in the GWAS (30,344 clinically diagnosed AD cases vs. 52,427 controls) performed by Kunkle *et al.* [2]. Twenty-one SNPs were associated with the risk of AD at genome-wide significance ($P$ value $5\times10^{-8}$). Summary statistics for 20 of these SNPs were available in the GWAS of

periodontitis, and so were included as instrumental variables for AD in our study. Summary statistics of periodontitis were obtained from the latest and largest periodontitis GWAS by Shungin et al. [13]. We only used data of people with European ancestry by excluding those with Hispanic/Latino background in the GLIDE consortium (12,289 clinically diagnosed periodontitis cases vs. 22,326 controls) (S2 Table).

### MR analysis

First, we applied MR-PRESSO (pleiotropy residual sum and outlier) to detect any horizontal pleiotropic outliers [14]. Pseudo $R^2$ that represents proportion of variance of liability explained by SNPs and F-statistic were calculated to evaluate the strength of the instruments. For appraising causality in both directions, the inverse-variance weighted (IVW) method (random effects) was used to calculate MR estimates [15], complemented with weighted median [16] and MR-Egger [17] methods that are relatively robust to horizontal pleiotropy, and with the MR-RAPS (robust adjusted profile score) method that is more robust to weak instrument bias [18]. To test for horizontal pleiotropy, we calculated the intercept and 95% confidence interval (CI) of the MR-Egger regression line [17]. We tested for heterogeneity between the causal estimates of individual SNPs using Cochran's Q statistic for the IVW and MR-Egger methods. Leave-one-out analyses were performed to ascertain that the effect was not disproportionately influenced by a single SNP. MR estimates are reported as an odds ratio (OR) for the outcome per unit increase in ln (OR) of the exposure. We emphasize that the calculation of MR estimate associated with a binary exposure (unlike a continuous exposure) is more efficient for identifying presence of a causal effect than quantifying the magnitude of the causal effect [19]. All statistical analyses were performed in R (version 3.6.1), with packages *TwoSampleMR* (0.4.25), *MRPRESSO* (1.0) and *meta* (4.9–7).

Ethical approval had been obtained in the original studies [2, 12, 13].

### Results

No outliers in the bidirectional MR analyses were detected with MR-PRESSO (global test $P>0.19$ for all). Using the five or the seven independent SNPs for periodontitis, pseudo $R^2$ value was 0.008 vs. 0.006 and F-statistic was 19.0 vs. 31.9 respectively, which suggests relatively weak instruments for periodontitis in this study. On the contrary, using the 20 SNPs as instruments for AD, the pseudo $R^2$ and F-statistic were 0.065 and 286.2 respectively.

The results of the bidirectional MR estimates are presented in Table 1. Using the five SNPs in the study of Munz et al. as periodontitis instruments, there was a weak association between

**Table 1. Bidirectional MR estimates for the association between chronic periodontitis and Alzheimer's disease.**

| Method | Periodontitis on Alzheimer's disease | | | | | | Alzheimer's disease[†] on periodontitis | | |
|---|---|---|---|---|---|---|---|---|---|
| | Instrumental SNPs[*] from Munz et al. | | | Instrumental SNPs[#] from Shungin et al. | | | | | |
| | OR (95% CI) | *P* value | Q statistic/ *P* value | OR (95% CI) | *P* value | Q statistic/ *P* value | OR (95% CI) | *P* value | Q statistic/ *P* value |
| IVW | 1.10 (1.02, 1.19) | 0.02 | 2.76/0.60 | 0.97 (0.87, 1.08) | 0.59 | 4.13/0.66 | 1.00 (0.96, 1.04) | 0.85 | 21.46/0.31 |
| Weighted median | 1.08 (0.98, 1.20) | 0.13 | | 0.99 (0.86, 1.14) | 0.89 | | 0.98 (0.94, 1.03) | 0.44 | |
| MR-Egger | 0.97 (0.68, 1.39) | 0.89 | 2.27/0.52 | 1.05 (0.86, 1.27) | 0.66 | 3.28/0.66 | 0.98 (0.93, 1.03) | 0.41 | 20.00/0.33 |
| MR-RAPS | 1.10 (1.01, 1.20) | 0.02 | | 0.97 (0.86, 1.09) | 0.60 | | 0.99 (0.95, 1.04) | 0.81 | |

CI: confidence interval; OR: odds ratio; IVW: inverse-variance weighted; MR: Mendelian randomization; RAPS: robust adjusted profile score; SNP: single-nucleotide polymorphism

[*]Five/seven[#] SNPs were used as instrumental variables for periodontitis respectively

[†]Twenty SNPs were used as instrumental variables for Alzheimer's disease

genetically predicted periodontitis and the risk of AD (OR 1.10, 95% CI 1.02 to 1.19, *P* = 0.02) based on the IVW method, which was supported by the MR-RAPS method (OR 1.10, 95% CI 1.01 to 1.20, *P* = 0.02). The MR estimate using the weighted median method was similar but with a wider 95% CI (OR 1.08, 0.98 to 1.20). The MR-Egger estimate was less consistent though (OR 0.97, 95% CI 0.68 to 1.39). However, the validation analysis using the seven non-overlapping SNPs in the study of Shungin *et al.* as periodontitis instruments did not support a causal effect of genetic liability to periodontitis on the risk of AD (Table 1). Using the 20 SNPs as instruments for AD, there was no association between genetically predicted AD and the risk of periodontitis (OR 1.00, 95% CI 0.96 to 1.04, *P* = 0.85). (Fig 1A to 1B) displays the MR estimates derived from the IVW method summarizing the effect from each individual SNP for periodontitis on the risk of AD, whilst Fig 2 shows the MR estimate derived from the IVW method summarizing the effect from each individual SNP for AD on the risk of periodontitis. No substantial evidence for horizontal pleiotropy was observed in the MR-Egger regression analyses (periodontitis on risk of AD using the five SNPs from the study of Munz *et al.*: intercept 0.027, 95% CI -0.050 to 0.101, *P* = 0.54; periodontitis on AD using the seven SNPs from

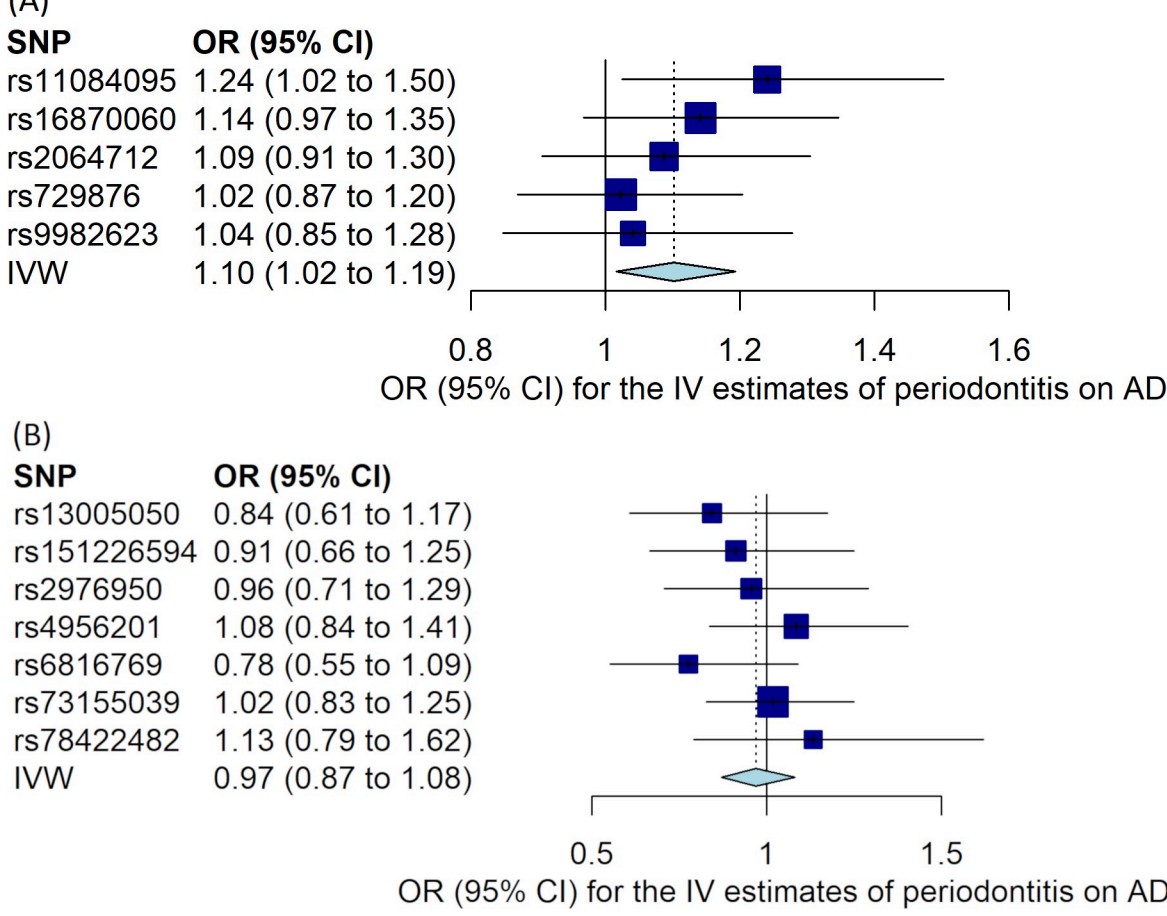

**Fig 1. Mendelian randomization (MR) estimates for the risk of Alzheimer's disease associated with periodontitis.** (A) Using instrumental SNPs from Munz *et al.* for periodontitis. (B) Using instrumental SNPs from Shungin *et al.* for periodontitis. MR estimates were calculated using the inverse-variance weighted (IVW) method to summarize the effect from each individual single-nucleotide polymorphism (SNP) in a random effects model. Odds ratio (OR) represents the risk of Alzheimer's disease per genetically determined 1-unit increase in ln (OR) of periodontitis. 95% CI: 95% confidence interval of the odds ratio. AD: Alzheimer's disease; IV: instrumental variable.

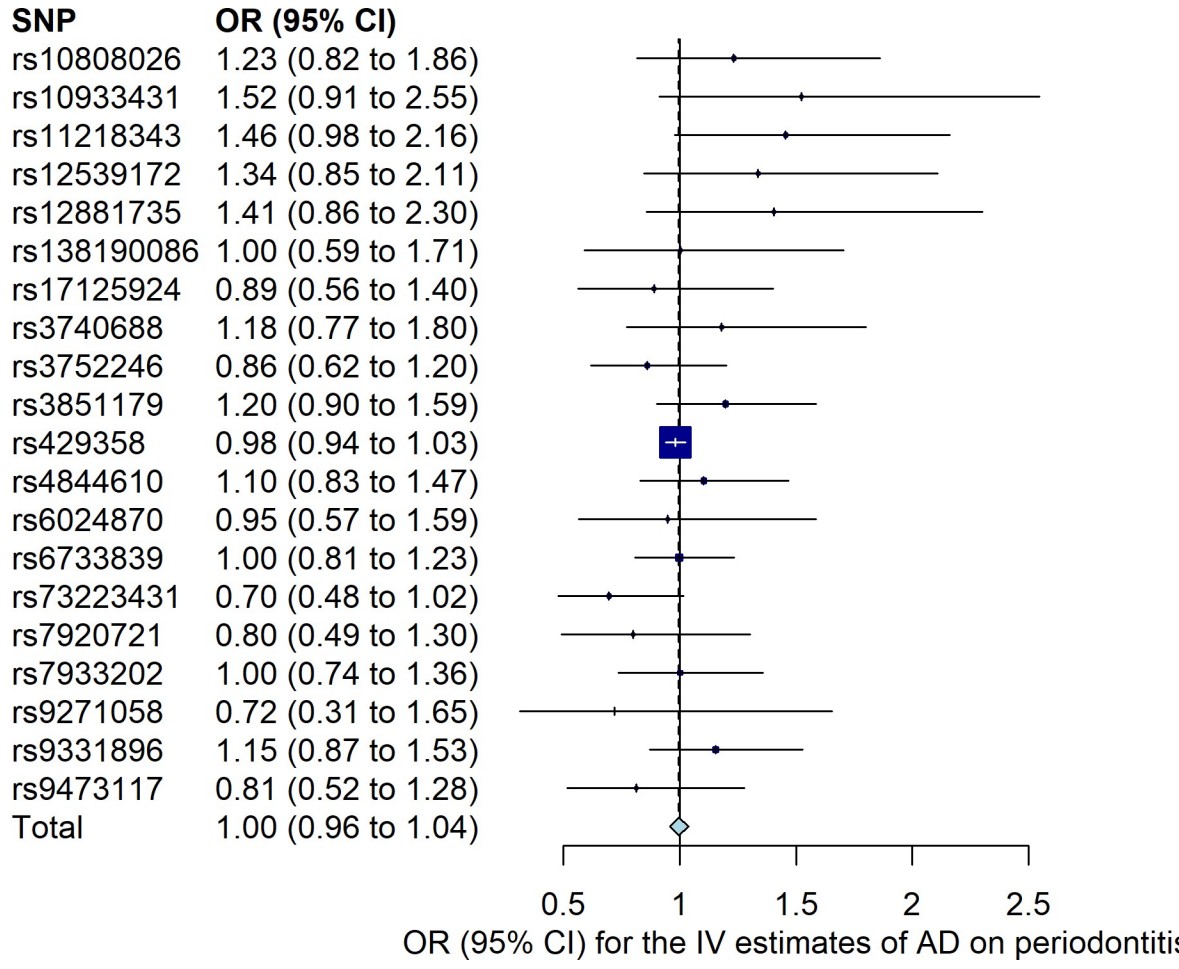

**Fig 2. Mendelian randomization (MR) estimate for the risk of periodontitis associated with Alzheimer's disease.** MR estimate was calculated using the inverse-variance weighted (IVW) method to summarize the effect from each individual single-nucleotide polymorphism (SNP) in a random effects model. Odds ratio (OR) represents the risk of periodontitis per genetically determined 1-unit increase in ln (OR) of Alzheimer's disease. 95% CI: 95% confidence interval of the odds ratio. AD: Alzheimer's disease; IV: instrumental variable.

the study of Shungin *et al.*: intercept -0.017, 95% CI -0.052 to 0.019, $P = 0.40$; and AD on risk of periodontitis: intercept 0.008, 95% CI -0.005 to 0.020, $P = 0.27$). There was no strong evidence for heterogeneity between SNPs evaluated by Cochran's Q statistic ($P > 0.31$ for all). The results from leave-one-out analyses did not suggest that the effects were disproportionately influenced by a single SNP except for the effect of rs429358 on risk of periodontitis (S1A to S1C Fig). The assumption of instrumental variables (the five and the seven SNPs for periodontitis) being independent of confounding factors was supported since genome-wide significant associations with other traits than periodontitis were not found by searching in a GWAS catalogue (https://www.ebi.ac.uk/gwas/).

## Discussion

Chronic periodontitis and infection with *P. gingivalis* were associated with cognitive impairment [9]. Periodontitis at baseline was associated with an increase in cognitive decline in patients with AD after six months of follow-up [20]. More recently, two independent groups

have shown that *P. gingivalis* and its virulence factor gingipains in the brain may play a central role in the pathogenesis of AD [7, 8]; repeated oral administration of *P. gingivalis* to mice resulted in neuroinflammation, neurodegeneration, and formation of amyloid plaque. In our study, a weak causal effect of genetic liability to periodontitis on the risk of AD derived from the primary analysis was not validated when we used the data from another newly published GWAS of periodontitis [13]. Overall, our study did not provide convincing evidence for periodontitis being a causal factor for the development of AD in the human setting.

Although we have used the available up-to-date data, this study has several potential limitations. MR reflects an average lifetime risk, so it cannot answer the question such as if having periodontitis during a certain period has any impact on the risk of AD. While our study had a sufficient power (with narrow 95% CIs) to investigate a potential causal effect of liability of AD on the risk of periodontitis, the power might not be sufficient to detect a small effect of periodontitis on the risk of AD. Although we used the two most recent GWAS of periodontitis in the population of European ancestry [12, 13], the weak instrument bias may still be an issue as indicated by the F-statistics and $R^2$ values. All SNPs used as instruments for periodontitis were weakly associated with periodontitis with a cut-off *P* value $5 \times 10^{-6}$ instead of $5 \times 10^{-8}$. Moreover, there was no overlapping SNPs between the two periodontitis GWAS, which further implies weak instruments for periodontitis. In addition, the biological mechanisms for most of these SNPs related to periodontitis are unclear. In two-sample MR any bias from weak instruments shifts the MR estimate towards the direction of the null [11], which may explain the unclear association between periodontitis and the AD risk we observed. It has been suggested that genetic predispositions are important for both the onset and the progression of periodontitis, and the heritability was estimated as high as 50% [21]. However, the GWAS of periodontitis up to date failed to identify consistent SNPs [12, 13, 22, 23]. The reason for divergent SNPs identified in periodontitis GWAS could be due to inconsistent definitions of periodontitis that were used in different studies. To confirm the causal effect of periodontitis on risk of AD, stronger instruments for periodontitis derived from large-scale GWAS with consistent definition of periodontitis [24] are warranted. Further MR studies using summary statistics from large GWAS of brain amyloid and tau deposition may examine the potential mechanisms between periodontitis and the risk of AD. It is also of interest to perform similar studies in other ethnic groups.

## Conclusions

The present study is the first to use a two-sample MR approach to investigate the causal relationship between periodontitis and AD bidirectionally in the population of European ancestry. By employing summary statistics from the latest and largest GWAS and various MR methods, we did not find convincing evidence to support periodontitis being a causal factor for the development of AD. There was also limited evidence to suggest genetic liability to AD being associated with the risk of periodontitis. Future GWAS of periodontitis with consistent definitions for this trait are needed to investigate the genetic role in periodontitis, and the genes identified from the high-quality GWAS can be further used to explore the potential causal role of periodontitis in various diseases.

## Supporting information

**S1 Table. Summary statistics for Mendelian randomization analysis of potential causal effect of periodontitis on Alzheimer's disease.**
(DOCX)

**S2 Table. Summary statistics for Mendelian randomization analysis of potential causal effect of Alzheimer's disease on periodontitis.**
(DOCX)

**S1 Fig. Leave-one-out Mendelian randomization (MR) estimates.** (A) Periodontitis on risk of Alzheimer's disease using instrumental SNPs from Munz *et al.* (B) Periodontitis on risk of Alzheimer's disease using instrumental SNPs from Shungin *et al.* (C) Alzheimer's disease on risk of periodontitis. MR estimates were calculated using the inverse-variance weighted (IVW) method in a random effects model after excluding each individual single-nucleotide polymorphism (SNP). The scale on x-axis represents ln [odds ratio (OR)] for the risk of the outcome per genetically determined 1-unit increase in ln (OR) of the exposure. The dots represent the MR estimates and the lines represent 95% confidence interval of the estimates. AD: Alzheimer's disease; MR: Mendelian randomization.
(TIF)

**S1 File. STROBE checklist.**
(DOC)

**S2 File. R code for the performed Mendelian randomization analyses.**
(R)

**S3 File. Data used in the R code.**
(XLSX)

## Author Contributions

**Conceptualization:** Yi-Qian Sun, Xiao-Mei Mai.

**Data curation:** Yi-Qian Sun, Rebecca C. Richmond, Xiao-Mei Mai.

**Formal analysis:** Yi-Qian Sun, Xiao-Mei Mai.

**Investigation:** Yue Chen, Xiao-Mei Mai.

**Methodology:** Yi-Qian Sun, Rebecca C. Richmond, Xiao-Mei Mai.

**Writing – original draft:** Yi-Qian Sun.

**Writing – review & editing:** Yi-Qian Sun, Rebecca C. Richmond, Yue Chen, Xiao-Mei Mai.

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
