## [Decision Letter · Decision Letter 0]

14 Nov 2019

PONE-D-19-17973

Relationship between periodontitis and Alzheimer’s disease: A bidirectional Mendelian randomization study

PLOS ONE

Dear Dr. Sun,

Thank you for submitting your manuscript to PLOS ONE, and my sincere apologies for the delay in the review process. After careful consideration, we feel that your work has merit but does not fully meet PLOS ONE’s publication criteria as it currently stands. Therefore, we invite you to submit a revised version of the manuscript that addresses the points raised during the review process.

Please pay particular attention to the reviewers' requests to use the most recent AD GWAS, report more methodological detail and make the code available.

We would appreciate receiving your revised manuscript by Dec 22 2019 11:59PM. To enhance the reproducibility of your results, we recommend that if applicable you deposit your laboratory protocols in protocols.io, where a protocol can be assigned its own identifier (DOI) such that it can be cited independently in the future. For instructions see: http://journals.plos.org/plosone/s/submission-guidelines#loc-laboratory-protocols

We look forward to receiving your revised manuscript.

Kind regards,

Kristel Sleegers

Academic Editor

PLOS ONE

Journal Requirements:

2. Your ethics statement must appear in the Methods section of your manuscript. If your ethics statement is written in any section besides the Methods, please move it to the Methods section and delete it from any other section. Please also ensure that your ethics statement is included in your manuscript, as the ethics section of your online submission will not be published alongside your manuscript.

Reviewers' comments:

Reviewer's Responses to Questions

**Comments to the Author**

1. Is the manuscript technically sound, and do the data support the conclusions?

Reviewer #1: No

Reviewer #2: Partly

2. Has the statistical analysis been performed appropriately and rigorously? 

Reviewer #1: Yes

Reviewer #2: No

3. Have the authors made all data underlying the findings in their manuscript fully available?

Reviewer #1: Yes

Reviewer #2: No

4. Is the manuscript presented in an intelligible fashion and written in standard English?

Reviewer #1: Yes

Reviewer #2: Yes

5. Review Comments to the Author

Reviewer #1: I recommend resubmitting with major revisions. First, the thoroughness of the analysis is commendable. The use of multiple mendelian randomization (MR) estimate methods, pleiotropy outlier measurements, reverse-causality, and leave-one-out analysis provide a better understanding of the results. However, the clumping threshold of the SNPs tested should be reported. This is to minimize the impact of linkage disequilibrium.

Unfortunately, the weakest parts of the manuscript are the data used for the analysis. First, the exposure data, the GWAS of periodontitis, only included five SNPs with relatively high P value threshold (5e-6). The norm is to use P value of 5e-8 and include more SNPs. I recommend calculating and reporting the F-statistic and the R2 value, which should explain the strength of the instrument and the variance of the SNPs. It would help assert the first assumption of MR: the instrumental variance is robustly associated with the exposure. The authors somewhat address this in the discussion ("…bias from weak instruments shifts the MR estimate towards the direction of the null…") but quantifying the instrument strength would put the results into greater context.

Similarly, the small GWAS study used for the reverse causation analysis (2,681 cases and 1,823 controls) may not provide sufficient power. As such, it is unclear that the result of the reverse causation analysis is due to lack of reverse causality or from lack of statistical power. The authors should provide the appropriate statistical power calculations to address this possibility.

Lastly, the outcome data, the GWAS of Alzheimers Disease (AD), is not the latest GWAS of the given disease and does not include the latest loci associated with AD. Latest meta-analysis GWAS has found 29 loci associated with AD (Jansen et al. 2019), while this manuscript refers to only nineteen loci found in an older GWAS. The summary statistics from the latest study are publicly available. The authors should run the analyses again, both forward and reverse causation analyses, using this dataset as their AD data.

In summary, I recommend major revision of the manuscript. Clumping threshold should be reported. F-statistics and R2 of all exposure instruments, as well as the power calculation for the reverse causation analysis should be calculated and reported. Finally, additional analysis using a more up-to-date AD GWAS summary data should be done.

Reviewer #2: Summary

Mendelian randomization (MR) is a strong tool for a better understanding of potential causal effects especially in disorders with a long-term duration such as AD. However, if not done properly, MR studies are prone to misinterpretation due to the complexities involved in the analysis. In this paper, the authors have looked at the relationship between AD and periodontitis. They have used summary stats from the related GWAS studies and concluded that there is no relationship. The authors have performed extensive analysis utilizing a number of MR techniques.

Issues

- First, the analysis is not public. In order to ensure reproducibility, it is crucial to make the code or analysis notebook publicly available. Especially with computational and data analysis work, unintentional coding mistakes might be made; availability of code/notebook helps with the review and future reproducibility.

- Another major issue with the paper is not using the latest Alzheimer GWAS. In the latest study, 29 risk-loci were identified https://www.nature.com/articles/s41588-018-0311-9

- Periodontitis SNPs are limited and very strong. Using a more relaxed p-value helps, 5e-8. Better to use R^2 for choosing the instrumental variable.

- MR analysis: authors have used multiple methods; however, some details are not properly reported such as SNPs clumping threshold.

6. PLOS authors have the option to publish the peer review history of their article (what does this mean?). If published, this will include your full peer review and any attached files.

Reviewer #1: Yes: Jonggeol Jeffrey Kim

Reviewer #2: No

---

## [Author Response · Author response to Decision Letter 0]

11 Dec 2019

The response letter to the reviewers are attached.

---

## [Decision Letter · Decision Letter 1]

9 Jan 2020

PONE-D-19-17973R1

Relationship between periodontitis and Alzheimer’s disease: A bidirectional Mendelian randomization study

PLOS ONE

Dear Dr. Sun,

Thank you for submitting your manuscript to PLOS ONE. We feel that the revisions have greatly improved the merit of the manuscript, and that it now meets PLOS ONE’s publication criteria. However, one of the reviewers recommended a small revision to the title that will indeed further improve the manuscript. Therefore, we invite you to submit a revised version of the manuscript with a changed title to reflect the null or mixed evidence for the relationship between periodontitis and Alzheimers Disease.

We would appreciate receiving your revised manuscript by Feb 23 2020 11:59PM. To enhance the reproducibility of your results, we recommend that if applicable you deposit your laboratory protocols in protocols.io, where a protocol can be assigned its own identifier (DOI) such that it can be cited independently in the future. For instructions see: http://journals.plos.org/plosone/s/submission-guidelines#loc-laboratory-protocols

We look forward to receiving your revised manuscript.

Kind regards,

Kristel Sleegers

Academic Editor

PLOS ONE

Reviewers' comments:

Reviewer's Responses to Questions

**Comments to the Author**

1. If the authors have adequately addressed your comments raised in a previous round of review and you feel that this manuscript is now acceptable for publication, you may indicate that here to bypass the “Comments to the Author” section, enter your conflict of interest statement in the “Confidential to Editor” section, and submit your "Accept" recommendation.

Reviewer #1: All comments have been addressed

Reviewer #2: All comments have been addressed

2. Is the manuscript technically sound, and do the data support the conclusions?

Reviewer #1: Yes

Reviewer #2: Yes

3. Has the statistical analysis been performed appropriately and rigorously? 

Reviewer #1: Yes

Reviewer #2: Yes

4. Have the authors made all data underlying the findings in their manuscript fully available?

Reviewer #1: Yes

Reviewer #2: Yes

5. Is the manuscript presented in an intelligible fashion and written in standard English?

Reviewer #1: (No Response)

Reviewer #2: Yes

6. Review Comments to the Author

Reviewer #1: I recommend resubmitting with minor revisions. I would like to thank the authors for the additional transparency, including the inclusion of F-statistics, clumping threshold, and the analysis code. While the periodontitis data, including the new data from Shungin et al., have fairly weak instrumental strength, quantifications of the instrumental strengths contextualize the mixed, and likely null, results. Additions to the Discussion and the Conclusion further contextualize and underline the limitations of the manuscript and what would be necessary for more robust future analyses. The title should echo this change and could be updated to reflect the null or mixed evidence for the relationship between periodontitis and Alzheimers Disease (AD).

I still believe that including AD GWAS with proxy cases (Janssen et al.) would be ideal but not necessary. AD has a strong heritability and the potential for introducing additional bias through the inclusion of proxy cases is minimal. However, larger GWAS size for AD will not solve the main issue of the analysis, which is the weak periodontitis data.

In summary, I recommend minor revision of the manuscript. Title should be changed to reflect the mixed/null finding of the manuscript. While not essential, I also recommend additional analysis using AD GWAS data with proxy cases.

Reviewer #2: Authors have addressed all the issues adequately. The availability of the analysis code will be helpful for future replication.

7. PLOS authors have the option to publish the peer review history of their article (what does this mean?). If published, this will include your full peer review and any attached files.

Reviewer #1: Yes: Jonggeol Jeff Kim

Reviewer #2: Yes: Faraz Faghri

---

## [Author Response · Author response to Decision Letter 1]

10 Jan 2020

Response to Reviewers is attached.

---

## [Editor Report · Decision Letter 2]

10 Jan 2020

Mixed evidence for the relationship between periodontitis and Alzheimer’s disease: A bidirectional Mendelian randomization study

PONE-D-19-17973R2

Dear Dr. Sun,

We are pleased to inform you that your manuscript has been judged scientifically suitable for publication and will be formally accepted for publication once it complies with all outstanding technical requirements.

With kind regards,

Kristel Sleegers

Academic Editor

PLOS ONE
---

## [Editor Report · Acceptance letter]

15 Jan 2020

PONE-D-19-17973R2 

Mixed evidence for the relationship between periodontitis and Alzheimer’s disease: A bidirectional Mendelian randomization study 

Dear Dr. Sun:

I am pleased to inform you that your manuscript has been deemed suitable for publication in PLOS ONE. Congratulations! Your manuscript is now with our production department. 

With kind regards,

on behalf of

Dr. Kristel Sleegers 

Academic Editor

PLOS ONE